# *Syncephalastrum massiliense* sp. nov. and *Syncephalastrum timoneanum* sp. nov. Isolated from Clinical Samples

**DOI:** 10.3390/jof10010064

**Published:** 2024-01-13

**Authors:** Jihane Kabtani, Fatima Boulanouar, Papa Mouhamadou Gaye, Muriel Militello, Stéphane Ranque

**Affiliations:** 1IHU Méditerranée Infection, 13005 Marseille, Francepapamg91@hotmail.com (P.M.G.);; 2MEPHI, SSA, IRD, AP-HM, Aix-Marseille Université, 13005 Marseille, France; 3VITROME, SSA, IRD, AP-HM, Aix-Marseille Université, 13005 Marseille, France

**Keywords:** mucormycosis, *Mucorales*, *Syncephalastrum*, genotype, phenotype, one new taxon

## Abstract

Mucormycosis is known to be a rare opportunistic infection caused by fungal organisms belonging to the *Mucorales* order, which includes the *Syncephalastrum* species. These moulds are rarely involved in clinical diseases and are generally seen as contaminants in clinical laboratories. However, in recent years, case reports of human infections due to *Syncephalastrum* have increased, especially in immunocompromised hosts. In this study, we described two new *Syncephalastrum* species, which were isolated from human nails and sputum samples from two different patients. We used several methods for genomic and phenotypic characterisation. The phenotypic analysis relied on the morphological features, analysed both by optical and scanning electron microscopy. We used matrix-assisted laser desorption–ionization time-of-flight mass spectrometry, energy-dispersive X-ray spectroscopy, and BiologTM technology to characterise the proteomic, chemical mapping, and carbon source assimilation profiles, respectively. The genomic analysis relied on a multilocus DNA sequence analysis of the rRNA internal transcribed spacers and D1/D2 large subunit domains, fragments of the translation elongation factor-1 alpha, and the β-tubulin genes. The two novel species in the genus *Syncephalastrum*, namely *S. massiliense* PMMF0073 and *S. timoneanum* PMMF0107, presented a similar morphology: irregular branched and aseptate hyphae with ribbon-like aspects and terminal vesicles at the apices all surrounded by cylindrical merosporangia. However, each species displayed distinct phenotypic and genotypic features. For example, *S. timoneanum* PMMF0107 was able to assimilate more carbon sources than *S. massiliense* PMMF0073, such as adonitol, α-methyl-D-glucoside, trehalose, turanose, succinic acid mono-methyl ester, and alaninamide. The polyphasic approach, combining the results of complementary phenotypic and genomic assays, was instrumental for describing and characterising these two new *Syncephalastrum* species.

## 1. Introduction

The *Syncephalastrum* species belongs to the *Mucorales* order [1]. These species are mostly found in tropical and subtropical areas of the environment in both the air and soil [2,3,4]. These are generally seen as clinical contaminants with a low pathogenicity and are rarely known to cause human diseases [5,6]. However, in recent years, case reports of human infections due to the *Syncephalastrum* genus have increased significantly, especially in immunocompromised hosts with diabetes [7,8], chronic hepatorenal disease [9], corneal infections, or those who have been the recipients of organ transplantations [4,10]. These human infections are usually related to the skin, nails, lungs, and central nervous system [11] and can have fatal outcomes, resulting in highly invasive diseases [12,13]. They can also cause chronic and acute infections in immunocompetent hosts [14,15].

Mucormycosisis a rare opportunistic infection caused by mucorales fungi, such as *Lichtheimia* (40%), *Rhizopus* (30%), *Syncephalastrum* (20%) and *Rhizomucor* (10%) [16]. This infection has become the third most common and fatal fungal infection after candidiasis and aspergillosis [10,17]. The most common species in the *Syncephalastrum* genus is *S. racemosum* [18,19,20]. The first report of a human infection caused by *Syncephalastrum* sp. was a cutaneous infection in an immunocompromised patient in India [20]. According to the Index Fungorum (www.indexfungorum.org, accessed on 17 October 2023), the *Syncephalastrum* genus is composed of two species: *S. racemosum* and *S. monosporum* (composed of three varieties, *S. monosporum* var. *monosporum*, *cristatum,* and *pluriproliferum*). Recently, the genus has been updated with six other species, namely *S. contaminatum*, *S. verruculosum*, *S. breviphorum*, *S. elongatum*, *S. simplex*, and *S. sympodiale* [21].

An accurate species morphological differentiation is considerably difficult to achieve using the usual mycological techniques in clinical laboratories. Different *Mucorales* genera, such as *Rhizomucor*, *Lichtheimia* and *Mucor*, have a laborious phenotypic identification and characterisation process due to their similar morphology. Moreover, the correct identification can only be achieved based on specific fungal structures. In practice, however, an adequate identification based on the morphological criteria is hampered by numerous exceptions. Among the *Mucorales* order, the *Cunninghamella* and *Syncephalastrum* genera can be easily distinguished. *Syncephalastraceae* fungi are typified by the production of cylindrical merosporangia on the surface of fertile vesicles [22].

The high inter- and intra-species phylogenetic diversity of the *Mucorales* order [15,23] is a real challenge for species identification and taxon delimitation. Numerous studies based on antifungal susceptibility tests and, more recently, on the use of molecular taxonomic methods, particularly the sequencing of the internal transcript spacers (ITSs) 1 and 2 and the D1/D2 domains of the large subunit (LSU) of the rRNA gene, displayed a higher efficiency for the identification of *Mucorales* fungi than morphology-based identification [22,24]. The reliable identification of these rare infection agents is the key to further strengthening our understanding of the species associated with epidemiology, pathogenicity, and their outcomes. This study aimed to analyse the phylogenetic status and describe the phenotypic characteristics of two new species of the *Syncephalastrum* genus, relying on multilocus sequence analysis and chemical and physiological characterisation.

## 2. Materials and Methods

### 2.1. Fungal Strains

Two novel *Syncephalastrum* strains were isolated from clinical samples from distinct patients at the University Hospital mycology laboratory in Marseille, France. *Syncephalastrum massiliense* PMMF0073 was isolated from a sputum sample from a 53-year-old patient diagnosed with HIV in 1988 and with syphilis in 2015. This patient had recently fractured and dislocated his elbow and had a radial head prosthesis. He had suffered from headaches, intermittent fever, and generalised rashes with scratching lesions for a few months. *Syncephalastrum timoneanum* PMMF0107 was isolated from the nails of a 38-year-old patient with a history of bronchiectasis, with no exacerbation of the disease. The two new isolates were deposited into two accessible culture collections, a BCCM/IHEM collection (IHEM 28561 and IHEM 28562) and a PMM (Parasitology Mycology Marseille) collection (PMMF0073 and PMMF0107), in an active form (conservation at 25 °C) and a lyophilised form (conservation at −80 °C).

### 2.2. MALDI-TOF MS Identification

The strains were inoculated in Petri dishes with Sabouraud dextrose agar (SDA) supplemented with gentamycin and chloramphenicol (GC) at 25 °C between four to seven days. After growth, we proceeded to identification using matrix-assisted laser desorption/ionization time-of-flight mass spectrometry (MALDI-TOF MS), following the protein extraction protocol described by Cassagne et al., 2016 [25]. The Microflex LT™ instrument, the MALDI Biotyper™ system (Bruker Daltonics GmbH, Bremen, Germany), and both the manufacturers’ databases as well as an in-house reference spectra database were used, as described in Normand et al., 2017 [26]. Moreover, the MALDI-TOF MS spectra of the two isolates were collected in addition to other spectra of the reference strains from the DSMZ and CBS collections, namely *S. racemosum* DSM 859, *S. monosporum* CBS 567.91, *S. monosporum* CBS 568.91, and *S. monosporum* CBS 569.91. All the spectra were used to construct a dendrogram based on the protein expression intensity using the MALDI-TOF Biotyper Compass Explorer (Bruker Daltonics).

### 2.3. DNA Extraction

DNA extraction was performed using the Qiagen™ Tissue Kit (Courtaboeuf, France) on the EZ1 Advanced XL(Qiagen) instrument. After five days of incubation at 25 °C on Sabouraud dextrose agar + gentamicin and chloramphenicol (SDA GC), a few colonies were picked from each sample and poured into bead tubes held in 600 μL of lysis buffer G2 (provided with the Qiagen™ Tissue Kit). The next step was mechanical lysis using the FastPrep™-24 instrument. One run was conducted at 6 m/second for 40 s, followed by a centrifugation at 10,000 rpm for one minute. Then, 200 μL of the supernatant was poured into a flat tube provided in the kit. The extraction finally began using the EZ1 Advanced XL instrument, according to the manufacturer’s instructions. The total elution volume of 100 μL of extracted genomic DNA was stored at −20 °C for further analysis.

### 2.4. DNA Amplification and Sequencing

Four genes were targeted, namely the internal transcribed spacers 1 and 2 (ITS1/ITS2) in the rRNA small subunit (SSU), a fragment of the β-tubulin gene (TUB2), a fragment of the translation elongation factor-1 alpha gene (TEF-1-α), and the D1/D2 domains of the rRNA large subunit (LSU) (Table 1).

For each gene, a PCR mix was prepared as follows. A total of 5 μL of the DNA extract was added to 20 μL of the mix (12.5 μL ATG (Ampli Taq Gold™ 360 Master Mix, Applied Biosystems™, Waltham, MA, USA)/6 μL sterile water DNase/RNase free/0.75 μL forward/reverse primer) to achieve a total volume of 25 μL per well. Particularly for the ITS gene, in order to ensure the entire sequence length, three PCR mixes were prepared for each sample with the ITS1/2, ITS3/4, and ITS1/4 primers amplifying the ITS1, ITS2, and ITS1-5.8S-ITS2 regions, respectively. The PCR programme for all the fungal gene amplifications was constituted by an initial denaturation step at 95 °C for 15 min, followed by 39 cycles at 95 °C for one minute of denaturation, 56 °C for 30 s of annealing, 72 °C for one minute of an extension step, and a final extension at 72 °C for five minutes. The PCR amplicons were revealed on a 2% agarose gel with the addition of the Sybr SafeTM DNA gel stain (Invitrogen). The gel was visualised using the Safe Imager 2.0 Blue-Light Transilluminator™ (Invitrogen). A total of 4 μL of the purified DNA was added to the BigDye™ mix (terminator cycle sequencing kit (Applied Biosystems), 1 μL BD/1.5 μL TP/3 μL sterile water DNase/RNase free/0.5 μL forward/reverse primer) to achieve a total volume of 10 μL per well. The sequencing reactions for all the genomic regions, consisting of 96 °C for one minute, followed by 25 cycles at 96 °C for 10 s, 50 °C for five seconds, and 60 °C for three minutes, were processed using a 3500 Genetic Analyzer™ (Applied Biosystems, Inc.). The sequences obtained were assembled and corrected using ChromasPro 2.0. All the sequences were deposited in GenBank, and the accession numbers are presented in Table 2.

### 2.5. Phylogenetic Analysis

In addition to the sequences of the six strains, we added 16 other reference strain sequences obtained from the GenBank database (the accession numbers are presented in Table 2). Two phylogenetic trees were constructed using the maximum parsimony (MP) method, the MEGA (Molecular Evolutionary Genetics Analysis) software version 11 [31] using the default settings, and 1000 bootstrap replications to assess the branch robustness. The first tree was based on the ITS sequences of all the strains and the second tree was based on the concatenated ITS, TUB2, TEF-1-α, and D1/D2 sequences of the six following strains: *Syncephalastrum massiliense* PMMF0073, *Syncephalastrum timoneanum* PMMF0107, *S. racemosum* DSM 859, *S. monosporum* CBS 567.91, *S. monosporum* CBS 568.91, and *S. monosporum* CBS 569.91. *Rhizopus microsporus* ATCC 52813 was used as an outgroup. A Bayesian phylogenetic inference was also achieved. Two other multi locus phylogenetic trees were constructed using the MrBayes software (3.2.7a) [32] and Figtree (V.1.4.4) [33].

### 2.6. Macroscopic Characterisation

To study their growth temperature profiles and macroscopic characters, such as the time of growth, colony morphology, and surface and reverse colours, the six strains were cultivated on SDA GC plates for seven days. They were then subcultured on other SDA GC plates, which were incubated at different temperatures, 4 °C, 25 °C, 30 °C, 37 °C, 40 °C, and 45 °C, and on a dehydrated medium (peptone: 5 g/L; glucose: 10 g/L; potassium dihydrogen phosphate: 1 g/L; magnesium sulphate: 0.5 g/L; dichloran: 0.002 g/L; chloramphenicol: 50 mg/L; agar: 15 g/L; pH: 5.6 ± 0.2) at 30 °C.

### 2.7. Microscopic Characterisation

To compare the microscopic features of the different fungal structures (hyphae, spores, and vesicles), fresh cultures of the six strains on SDA GC plates were first examined using optical microscopy. The slides were prepared by gently dabbing the surface of the fungal colony with adhesive tape. The tape was then mounted with one drop of lactophenol cotton blue between the slide and the slipcover. Photographs were taken using a DM 2500 (Leica Camera SARL, Paris, France).

Scanning electron microscopy (SEM) was performed using the TM4000 Plus (Hitachi High-Technologies, Tokyo, Japan) microscope via the 15 KeV lens mode 4 with a backscattered electron detector. A fungal colony sample was cut from the Petri dish and placed on a microscopy slide. A volume of 400 μL of 2.5% glutaraldehyde in a 0.1 M sodium cacodylate buffer was poured over the fungal cut for fixation and stored at 30 °C until completely dry. The standardised fungal structures (hyphae, vesicle, sporangiola, merosporangium, and the number of sporangiospores within the merosporangial sack) were measured using a specific tool for the distance measurement included in the TM4000 Plus microscope. The results were represented in a principal component analysis (PCA) computed using the XLSTAT (Addinsoft, Paris, France) software V.2022.4.1.

### 2.8. Physiological Analysis

#### 2.8.1. EDX (Energy-Dispersive X-ray Spectroscopy)

Fresh colonies of the six strains were fixed for at least one hour with glutaraldehyde 2.5% in a 0.1 M sodium cacodylate buffer. Cytospin was performed using a volume of 200 μL from the fixed solution, followed by centrifugation at 800 rpm for eight minutes. EDX was carried out using an INCA X-Stream-2 detector (Oxford Instruments, Abingdon, UK) linked to the TM4000 Plus SEM and AztecOne software (Oxford instruments, UK). The slide chemical mapping was performed blindly, and all the chemical elements were taken into account. The weight and atomic percentages were subjected to a PCA computed using the XLSTAT (Addinsoft) software V.2022.4.1.

#### 2.8.2. Biolog™ Phenotypic Analysis

The phenotypic analysis was achieved using Biolog™ advanced phenotypic technology, as previously used for yeast characterisation by Kabtani et al. [34]. This system characterises microorganisms using a patented Redox tetrazolium dye that changes colour in response to cellular respiration in 96-micro-well plates and confers a metabolic fingerprint. We used the FF (filamentous fungi) MicroPlates (Gen III), for carbon source utilisation. The carbon sources were selected for their high discrimination between the fungal phenotype profiles [35]. All the wells contained the substrate and the dye, with the exception of the control well that only contained the dye. The strains were first cultivated on a malt extract agar (MEA) 2% medium and prepared with 20 g/L of malt extract and 18 g/L of agar in distilled water [36]. The fungal incubation time depended on its specific growth rate. The *Syncephalastrum* was a fast-growing genus that reached its maximal growth after five to seven days. After the colonies had developed and the hyphae colour had turned from white to dark brown, the fungal suspension was prepared in the FF inoculating fluid (Biolog part number 72106) by swabbing the surface of the colony. The transmittance levels were adjusted between 75% and 80% using a Biolog™ Turbidimeter [35]. The assay was performed in triplicate and 100 μL of the suspension was poured into each well of the FF MicroPlates (Biolog part number 1006), which were incubated at 26 °C for seven days and read using the Biolog MicroStation™ Reader (Biolog, Inc, Hayward, CA, USA). The results were represented as a heat map, performed using the XLSTAT™ (Addinsoft) software V.2022.4.1.

### 2.9. Antifungal Susceptibility Testing (AFST)

We determined the in vitro activity of ten antifungal drugs, namely amphotericin B, voriconazole, posaconazole, itraconazole, isavuconazole, fluconazole, micafungin, anidulafungin, flucytosine, and caspofungin, against the two clinical isolates and the four type strains from the *Syncephalastrum* genus. The minimal inhibitory concentration of each antifungal was determined using the E-test™ (bioMérieux, Craponne, France) concentration gradient agar diffusion assay, as described in Kondori et al., 2011 [37].

## 3. Results

### 3.1. MALDI-TOF MS Identification

The MALDI-TOF MS identification of the new isolates, *Syncephalastrum massiliense* PMMF0073 and *Syncephalastrum timoneanum* PMMF0107, did not match any spectrum present in our laboratory database. The strains spectra did, however, reveal pertinent information about protein expression profiles that was interesting for strain differentiation. The dendrogram (Figure 1) revealed the similarity of each isolate with a distinct *Syncephalastrum* species. *Syncephalastrum timoneanum* PMMF0107 clustered with *S. racemosum* DSM 859 and *Syncephalastrum massiliense* PMMF0073 clustered with the *S. monosporum* clade.

### 3.2. DNA Sequencing and Phylogenetic Analysis

The ITS region was recognised as the most precise and distinct marker in the *Mucorales* order (Ramesh et al., 2010) [13]. However, the two isolate sequences (the accession numbers are provided in Table 2) queried using the search tool (BLAST/NCBI) (http://blast.ncbi.nlm.nih.gov/blast, accessed on 17 October 2023) against the NCBI nucleotide database showed a less than 98% identity with the available nucleotide sequences, which was below the usual species identification threshold. Four dendrograms were built. The first was based on the ITS sequences of 22 strains. The second was based on the concatenation of four loci (ITS, TUB2, TEF1 and D1/D2) from six strains. In the first trees (Figure 2 and Figure 3), each new isolate clustered with a distinct clade. *Syncephalastrum massiliense* PMMF0073 appeared closely related to *S. racemosum*, while *Syncephalastrum timoneanum* PMMF0107 appeared relatively distant from both *S. racemosum* and *S. monosporum.* Furthermore, the second trees (Figure 4 and Figure 5) illustrated the distinct genomic features of the two novel species, which were relatively distant from one another, each clustering with a distinct *Syncephalastrum* species: *S. timoneanum* PMMF0107 with *S. racemosum* and *S. massiliense* PMMF0073 with *S. monosporum.*

### 3.3. Macroscopic Characterisation

The macroscopic morphological features of the six strains showed a rapid time of growth on the SDA GC medium, with an optimal growth temperature of 25 °C. Colonies with a fluffy and cottony aspect appeared after two to three days of incubation. The colour of the mycelium was white after 48 h, then became darker after 72 h, and reached a high level of sporulation around day five. The mycelia of *Syncephalastrum massiliense* PMMF0073, *Syncephalastrum timoneanum* PMMF0107, and *S. monosporum* CBS 567.91 were dark in colour, while they were grey for *S. monosporum* CBS 568.91 and *S. monosporum* CBS 569.91. *S. racemosum* DSM 859 displayed a lighter colour. All the isolates were xerotolerant as they grew on a dehydrated medium. None of them grew at 4 °C, 40 °C, or 45 °C (Figure 6).

### 3.4. Microscopic Characterisation

The colonies on the SDA of *S. massiliense* PMMF0073 and *S. timoneanum* PMMF0107 at 25 °C after 5 days were fluffy and cottony. The mycelium was initially white, then became darker with age.

Microscopic observations revealed, for both species, irregularly branched wide and aseptate hyphae with a ribbon-like aspect. Rhizoids and stolons were not observed. The sporangiophores were derived from aerial hyphae that were straight, lightly bent, single-branched, or unbranched (3–13 μm in wide). Terminal vesicle ovoid and globose were present at the apices for all the strains with different lengths. Depending on the species, the terminal vesicle generated cylindrical merosporangia over the whole surface, containing several merospores in a single row. The absence of chlamydospores and zygospores was unknown.

The *S. monosporum* species presented the largest hyphae (13–17 μm) and smallest vesicles (15–28 μm) in comparison with *Syncephalastrum massiliense* PMMF0073 (Figure 7 and Figure 10), *Syncephalastrum timoneanum* PMMF0107 (Figure 8 and Figure 11), and *S. racemosum* DSMZ 859 (Figure 9 and Figure 12), which, in contrast, displayed smaller hyphae (7–13 μm) and larger vesicles (29–31 μm). The surface of the *S. monosporum* vesicle was entirely covered by sporangiola (4–7 μm) (Figure 9 and Figure 12). However, the vesicle surfaces of *Syncephalastrum massiliense* PMMF0073, *Syncephalastrum timoneanum* PMMF0107, and *S. racemosum* DSMZ 859 were all surrounded by grey cylindrical merosporangia (15–16 μm). Each merosporangial sack contained six or seven light grey merospores, which were smooth-walled and spherical to ovoid (3–6 μm). A PCA based on the fungal structure measures showed that the microscopic features of the two new strains were relatively similar to *S. racemosum* (Figure 13).
Figure 7Lactophenol cotton blue mount of *Syncephalastrum massiliense* PMMF0073. (**A**) Sporangiophore with apical vesicles and merosporangial sacks enclosing merospores. (**B**) Columella and hyphae ribbon-like aspect. (**C**) Merospores. Optical microscopy (magnification ×1000). Scale bars: 50 μm.
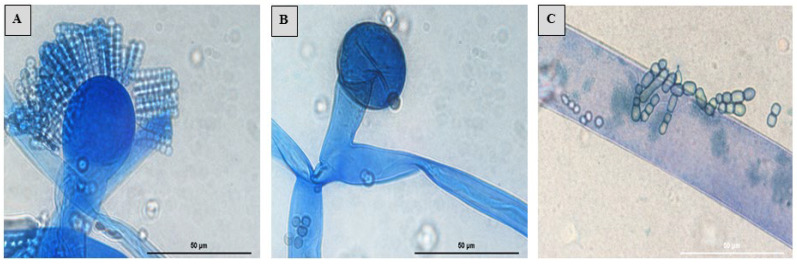

Figure 8Lactophenol cotton blue mount of *Syncephalastrum timoneanum* PMMF0107. (**A**) Sporangiophore with apical vesicles and merosporangial sacks enclosing merospores. (**B**) Columella and hyphae ribbon-like aspect. (**C**) Merospores. Optical microscopy (magnification ×1000). Scale bars: 50 μm.
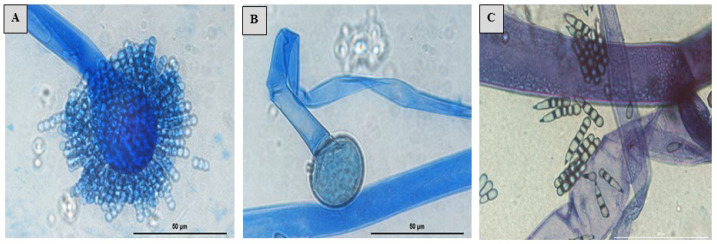

Figure 9Morphology of *Syncephalastrum* spp. (**A**) Sporangiophore with apical vesicles and merosporangial sacks of *Syncephalastrum racemosum* DSM 859. (**B**–**D**) Sporangiophore with apical vesicles and merosporangia of *S. monosporum* CBS 567.91, *S. monosporum* CBS 568.91, and *S. monosporum* CBS 569.91. (**E**,**F**) Columella of *S. racemosum* DSM 859 and *S. monosporum* CBS 567.91. Optical microscopy (magnification ×1000). Scale bars: 50 μm.
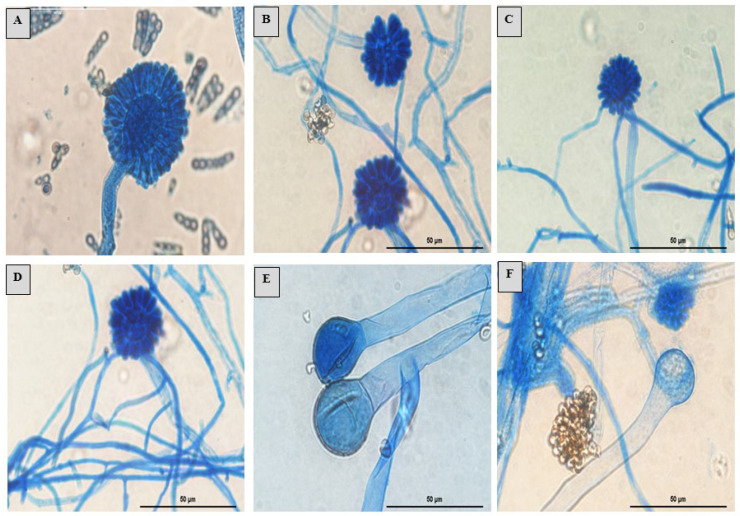

Figure 10Vesicles, sporangiophores, merosporangia, and merospores of *Syncephalastrum massiliense* PMMF0073. Scanning electron microscopy TM 4000Plus (15 KeV lens mode 4). Scale bars: (**A**) = 50 μm, (**B**) = 30 μm, (**C**) = 200 μm, and (**D**) = 40 μm.
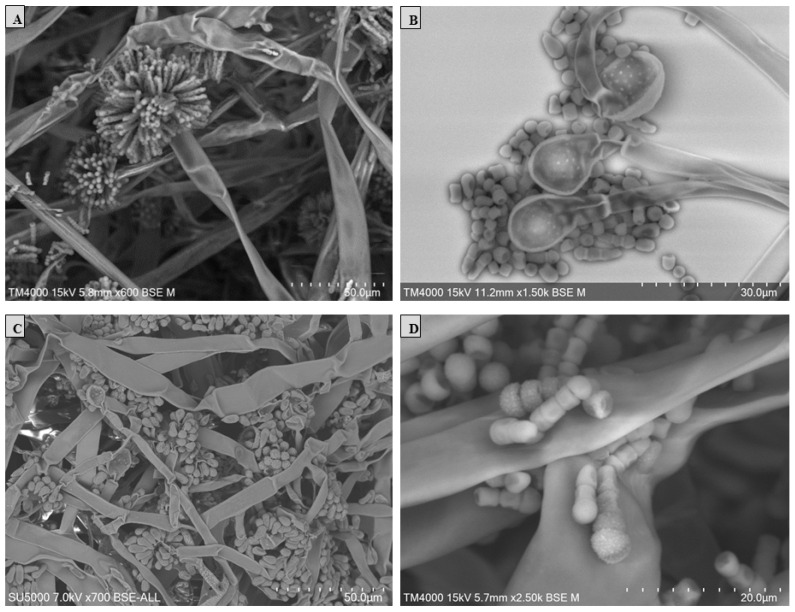

Figure 11Vesicles, sporangiophores, merosporangia, and merospores of *Syncephalastrum timoneanum* PMMF0107. Scanning electron microscopy TM 4000Plus (15 KeV lens mode 4). Scale bars: (**A**) = 50 μm, (**B**) = 30 μm, (**C**) = 50 μm, and (**D**) = 20 μm.
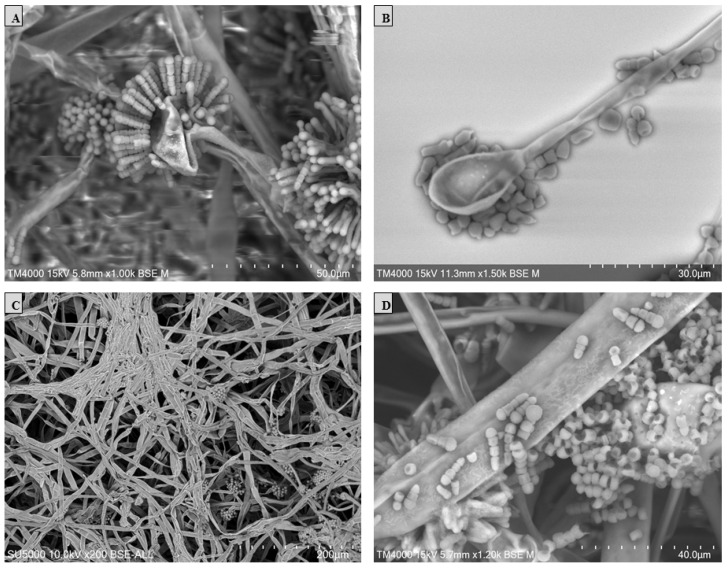

Figure 12Morphology of *Syncephalastrum* spp. (**A**) Sporangiophore with apical vesicles and merosporangial sacks of *S. racemosum* DSM 859. (**B**–**D**) Sporangiophore with apical vesicles and merosporangia of *S. monosporum* CBS 567.91, *S. monosporum* CBS 568.91, and CBS 569.91, respectively. Scanning electron microscopy TM 4000Plus (15 KeV lens mode 4). Scale bars: (**A**) = 40 μm, (**B**,**C**) = 30 μm, and (**D**) = 50 μm.
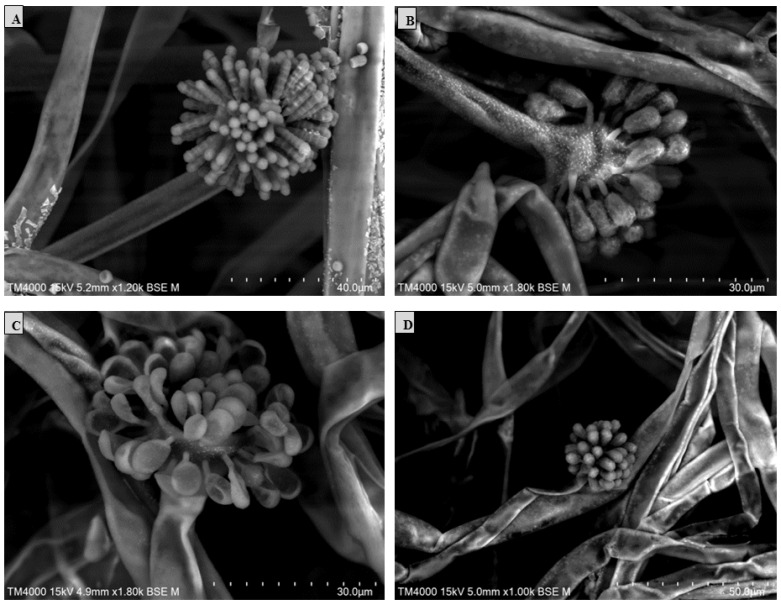

Figure 13Principal component analysis (PCA) of the different structure measurements (hyphae, columella, sporangiola, merosporangium, and sporangiospores number within the merosporangial sack) using the TM4000 Plus microscope (SEM) for the four reference strains and two new species of *Syncephalastrum*. In this analysis, computed using the XLSTAT software V.2022.4.1, the principal components F1 and F2 explained 90.3% of the fungi structure variance.
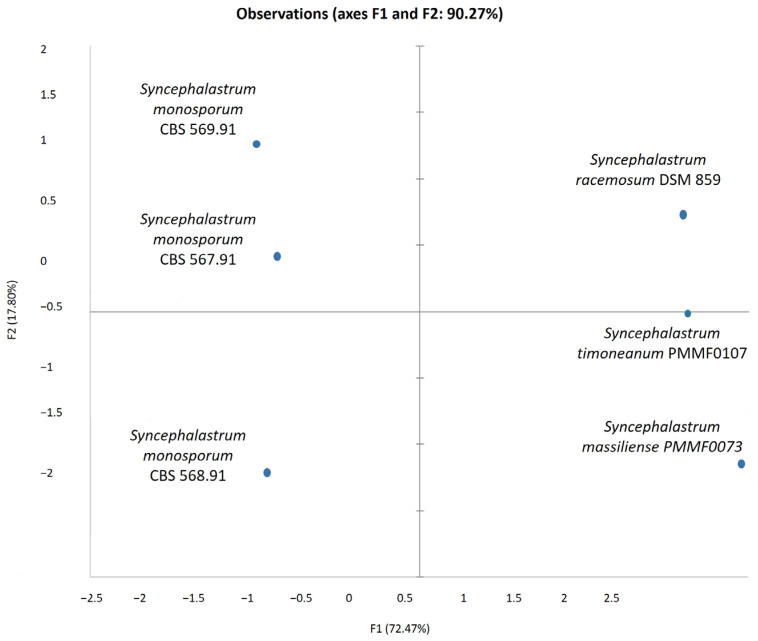


### 3.5. Physiological Analysis

#### 3.5.1. EDX (Energy-Dispersive X-ray Spectroscopy)

The results were represented as a PCA (Figure 14), showing that the chemical mapping profiles of the two new species differed from those of *S. racemosum* DSM 859. *Syncephalastrum timoneanum* PMMF0107 clustered with *S. monosporum* CBS 567.91, while *Syncephalastrum massiliense* PMMF0073 clustered with both *S. monosporum* CBS 568.91 and *S. monosporum* CBS 569.91.

#### 3.5.2. Biolog™ System

The Biolog™ phenotypic technology provided valuable information about the strain properties using a specific and precise microbial phenotypic characterisation. When reduction occurred in the FF plates, the dye colour changed to purple. The Biolog Omnilog equipment analysed images taken over time using a colour camera to quantify the reduced dye (Bochner et al., 2001). The results were represented as a heat map (Figure 15), which appeared fairly heterogeneous, showing that each new species had a substrate assimilation profile close to a distinct *Syncephalastrum* species. Most of the substrates were assimilated. The few non assimilated substrates were instrumental for species discrimination. *Syncephalastrum timoneanum* PMMF0107 appeared relatively close to *S. monosporum*. In contrast, *Syncephalastrum massiliense* PMMF0073 appeared closer to *S. racemosum*.

### 3.6. Antifungal Susceptibility Testing (AFST)

The minimal inhibitory concentrations (MICs) of the ten antifungal drugs evaluated are displayed in Table 3. All the strains exhibited high micafungin, anidulafungin, caspofungin, flucytosine, fluconazole, voriconazole, and isavuconazole MICs. The amphotericin B and itraconazole MICs were relatively low against *S. massiliense* PMMF0073, *S. timoneanum* PMMF0107, and *S. racemosum* DSM 859, whereas the posaconazole MICs were only low against *S. timoneanum* PMMF0107 and *S. racemosum* DSM 859. It was noteworthy that the itraconazole and posaconazole MICs were lower against *S. timoneanum* PMMF0107 and *S. racemosum* DSM 859 than against *S. massiliense* PMMF0073.

### 3.7. Taxonomy

*Syncephalastrum massiliense* Kabtani J. and Ranque S. sp. nov.

MycoBank: MB843858

(Figure 7 and Figure 10)

Etymology: Named after Marseille, the city where it was isolated.

Diagnosis: *Syncephalastrum massiliense* PMMF0073 was closely similar to *S. racemosum* DSMZ 859, based on the microscopic characteristics. Both species presented small hyphae (7–11 μm) and large vesicles (29–31 μm) in contrast to *S. monosporum,* which presented larger hyphae (13–17 μm) and smaller vesicles (15–28 μm). Moreover, the *S. monosporum* vesicle surface was entirely covered by sporangiola (4–7 μm). Meanwhile, both the *S. massiliense* PMMF0073 and *S. racemosum* DSMZ 859 vesicle surfaces were surrounded by merosporangia (15–16 μm). Each merosporangial sack contained six or seven merospores.

Type: France: Marseille. Human sputum, 11 September 2019. (PMMF0073—holotype; IHEM 28561—isotype). GenBank: OL699905 (ITS), ON149883 (Btub2), OM362516 (TEF-1a), OM417069 (D1/D2).

Description

Macromorphology: The macroscopic features showed that *Syncephalastrum massiliense* had a rapid growth time on the SDA GC medium, with an optimal temperature of growth at 25 °C. The strain was xerotolerant and growth was inhibited at temperatures ≤4 °C or ≥40 °C. Colonies with fluffy and cottony aspect were seen from two to three days post-inoculation. The mycelium was white at 48 h, then became darker at 72 h, and reached a high sporulation level around day five.

Micromorphology: The colonies on the SDA of *S. massiliense* PMMF0073 at 25 °C after 5 days were fluffy and cottony. The mycelium was initially white, then became darker with age. Microscopic observations revealed irregularly branched wide and aseptate hyphae with a ribbon-like aspect. Rhizoids and stolons were not observed. The sporangiophores were derived from aerial hyphae, which were straight, lightly bent, single-branched, or unbranched (3 –13 μm in wide). Terminal vesicle ovoid and globose were present at the apices. The absence of chlamydospores and zygospores was unknown. *Syncephalastrum massiliense* PMMF0073 displayed small hyphae (7–13 μm) and large vesicles (29–31 μm). The surfaces of the vesicles were all surrounded by grey cylindrical merosporangia (15–16 μm). Each merosporangial sack contained six or seven light grey merospores, which were smooth-walled and spherical to ovoid (3–6 μm).

The Biolog™ phenotypic technology provided information on the assimilation capacity of the fungus carbon sources. *S. massiliense* PMMF0073 was the only *Syncephalastrum* tested that assimilated the fewest substrates, including adonitol, alpha-methyl-D-glucoside, trehalose, turanose, succinic acid mono-methyl ester, and alaninamide. It was noteworthy that D-tagatose was only assimilated by *S. massiliense* PMMF0073 and *S. monosporum* CBS 567.91. *S. massiliense* PMMF0073 displayed a carbon source assimilation profile relatively similar to *S. racemosum* DSM 859.

### 3.8. Host: Human

*Syncephalastrum timoneanum* Kabtani J. and Ranque S. sp. nov.

MycoBank: MB843870

(Figure 8 and Figure 11)

Etymology: Named after La Timone, the hospital where it was isolated in Marseille, France.

Diagnosis: *Syncephalastrum timoneanum* PMMF0107 was closely related to *S. racemosum* DSMZ 859, relying on the microscopic characteristics. The two species presented small hyphae (7–11 μm) and large vesicles (29–31 μm) in contrast to *S. monosporum* species, which had larger hyphae (13–17 μm) and smaller vesicles (15–28 μm). In addition, while the vesicle surfaces of *S. monosporum* were fully covered by sporangiola (4–7 μm), the vesicle surfaces of both *S. timoneanum* PMMF0107 and *S. racemosum* DSMZ 859 were surrounded by merosporangia (15–16 μm). Each merosporangial sack contained six or seven sporangiospores (merospores).

Type: France: Marseille. Human nails, 02 March 2020. (PMMF0107—holotype; IHEM 28562—isotype). GenBank: OL699906 (ITS), ON149884 (Btub2), OM362517 (TEF-1a), OM417070 (D1/D2).

Description

Macromorphology: The macroscopic features revealed that *Syncephalastrum timoneanum* PMMF0107 had a rapid growth time on the SDA GC medium, with an optimal temperature of growth at 25 °C. The strain could grow on a dehydrated medium, demonstrating that was xerotolerant. However, no growth was observed at 4 °C, 40 °C, and 45 °C. Colonies with a fluffy and cottony aspect were seen from two to three days. The colour of the mycelium was white in the first 48 h, then became darker at 72 h, and reached a high level of sporulation around day five.

Micromorphology: The colonies on the SDA of *S. timoneanum* PMMF0107 at 25 °C for 5 days were fluffy and cottony. The mycelium was initially white, then became darker with age.

Microscopic observations revealed irregularly branched wide and aseptate hyphae with a ribbon-like aspect. Rhizoids and stolons were not observed. The sporangiophores were derived from aerial hyphae, which were straight, lightly bent, single-branched, or unbranched (3–13 μm in wide). Terminal vesicle ovoid and globose were present at the apices. The absence of chlamydospores and zygospores was unknown. *Syncephalastrum timoneanum* PMMF0107 presented small hyphae (7–13 μm) and large vesicles (29–31 μm). The surfaces of the vesicles were all surrounded by grey cylindrical merosporangia (15–16 μm). Each merosporangial sack contained six or seven light grey merospores, which were smooth-walled and spherical to ovoid (3–6 μm).

The Biolog™ advanced phenotypic technology provided information on the assimilation capacity of the fungus carbon sources. *S. timoneanum* PMMF0107 was the only species of the *Syncephalastrum* genus that assimilated most of substrates. However, there were some exceptional substrates that were not assimilated (D-tagatose, D-psicose, N-acetyl-D-mannosamine, L-fucose, glucuronamide, and sedoheptulosan). Based on this carbon source assimilation, *S. timoneanum* PMMF0107 displayed a relatively similar profile to the *S. monosporum* species.

### 3.9. Host: Human

Additional specimen examined (1): Type: Country of origin unknown. Before 24 January 1977. (DSM 859—holotype; ATCC 18192—isotype). GenBank: OL699907 (ITS), ON149885 (Btub2), OM362518 (TEF-1a), OM417071 (D1/D2).

Additional specimen examined (2): Type: China: Zhejiang Prov., Wuxing. Soil, 21 October 1960. (CBS 567.91—holotype). GenBank: OL699908 (ITS), ON149886 (Btub2), OM362519 (TEF-1a), OM417072 (D1/D2).

Additional specimen examined (3): Type: China: Jiangsu Prov., Nanjing. Soil, 13 October 1960. (CBS 568.91—holotype). GenBank: OL699909 (ITS), ON149887 (Btub2), OM362520 (TEF-1a), OM417073 (D1/D2).

Additional specimen examined (4): Type: China: Zhejiang Prov., Hangzhou. Pit mud, 19 October 1960. (CBS 569.91—holotype). GenBank: OL6999010 (ITS), ON149888 (Btub2), OM362521 (TEF-1a), OM417074 (D1/D2).

## 4. Discussion

According to Vu et al., 2019 [38], the two nuclear ribosomal sequences of the internal transcribed spacers (ITSs) and the D1/D2 domain of the large subunit (LSU) remain the most reliable genetic markers for establishing the taxonomic thresholds for filamentous fungal identification. The thresholds defined for fungi delimitation at the genus level were 94.3% based on the ITS barcodes and 98.2% based on the LSU barcodes. The best thresholds for discriminating filamentous fungi at the species level were predicted to be 99.6% for the ITS and 99.8% for the LSU. In this study, the BLASTn query for the two newly isolated species showed a ≤98% identity.

Thus, we proposed that *Syncephalastrum massiliense* PMMF0073, isolated from human sputum, and *Syncephalastrum timoneanum* PMMF0107, isolated from human nails, were two novel species in the *Syncephalastrum* genus based on their comprehensive phenotypic and genotypic analyses. The phenotypic analysis highlighted the distinct protein expression profiles of these two isolates, assessed using MALDI-TOF MS. Each one appeared closer to a different species of the *Syncephalastrum* genus. *Syncephalastrum timoneanum* PMMF0107 seemed closer to *S. racemosum* DSM 859 and *Syncephalastrum massiliense* PMMF0073 was closer to *S. monosporum* clade. The phylogenetic tree constructed using the four loci was congruent with the MALDI-TOF MS dendrogram and showed the same species clustering.

All the strains shared the following macroscopic features: the colony texture, time, and temperature of growth, as previously described [8,39]. Moreover, the mycelium colour of the new isolates was akin to *S. monosporum*. All the strains did not grow ≥40 °C. In contrast to our observations, some authors have described *S. racemosum* and *S. monosporum* as hydrophilic and thermotolerant moulds [12] or have declared that *Syncephalastrum* species were able to grow above 40 °C [6,10].

*S. racemosum* can be misidentified and confused with some black *Aspergillus* species, such as *Aspergillus niger* [5]. The hyphal morphology and the merosporangial sack enclosing sporangiospores are key for differentiating these two fungi, but also for distinguishing between the *Syncephalastrum* species. According to Hoffman et al., 2013 [40], *Syncephalastrum* is the only genus in the *Mucorales* order which produces merosporangia with merospores arranged in linear chains. Benjamin [22] also reported that *S. racemosum* produces sporangiospores in deciduous, tubular merosporangial sacks developed across the entire surface of an apical, spherical swelling of the sporangiophore. Indeed, the microscopic features are very useful for these fungi classifications. In fact, the two new isolated strains shared many *S. racemosum* morphological features, mainly relying on the number of sporangiospores contained in each merosporangial sack. *S. racemosum* merosporangium contained approx. six or seven sporangiospores, while *S. monosporum* contained only one sporangiola. In all the strains, the hyphae were large and aseptate with a ribbon-like aspect, as described by Gomes et al., 2011 [10]. Several comprehensive studies [41,42,43] based on *Mucorales* antifungal susceptibility testing reported significant variations between the genera, species, and strains within the *zygomycetes* class. The three species of *S. massiliense* PMMF0073, *S. timoneanum* PMMF0107, and *S. racemosum* DSM 859 were uniformly susceptible to amphotericin B and itraconazole. However, only *S. timoneanum* PMMF0107 and *S. racemosum* DSM 859 were susceptible to posaconazole. The susceptibility displayed by *S. racemosum* against the three antifungal drugs was reported by Vitale et al., 2012 [44]. In line with the microscopic analyses, the antifungal susceptibility profiles of the two novel species, *S. timoneanum* PMMF0107 and *S. massiliense* PMMF0073, were relatively closer to *S. racemosum* than to *S. monosporum.*

While the molecular methods and phenotypic methods, such as MALDI-TOF MS and morphological analysis, supplied no information about the strain properties, the Biolog™ system provided information on the assimilation capacity of the fungus carbon sources. Whereas the majority of the substrates were assimilated by all the strains, some relevant differences were helpful for discriminating between the two isolates. *S. massiliense* PMMF0073 was the strain that assimilated the fewest substrates. Among the substrates assimilated by all, except for *S. massiliense* PMMF0073, were adonitol, α-methyl-D-glucoside, trehalose, turanose, succinic acid mono-methyl ester, and alaninamide. One exception was D-tagatose, which was assimilated by *S. massiliense* PMMF0073 but not by *S. timoneanum* PMMF0107. In contrast, *S. timoneanum* PMMF0107 was the only species that assimilated almost all the substrates, except for D-tagatose, D-psicose, N-acetyl-D-mannosamine, L-fucose, glucuronamide, and sedoheptulosan. On the basis of these carbon source assimilation profiles, *S. timoneanum* PMMF0107 was close to the *S. monosporum* species and *S. massiliense* PMMF0073 was close to *S. racemosum* DSM 859. Additionally, relying on EDX chemical mapping, the new strains were fairly similar to the *S. monosporum* species. Finally, the morphological features, antifungal susceptibility tests, and the ITS and D1D2 tree highlighted the similarities of both *S. massiliense* and *S. timoneanum* with *S. racemosum*.

## 5. Conclusions

The two novel species in the genus *Syncephalastrum*, *S. massiliense* PMMF0073 and *S. timoneanum* PMMF0107, had a similar morphology to *S. racemosum*. However, each displayed distinct phenotypic and genotypic features. The polyphasic approach, combining the results of complementary assays, including the Biolog™ system, MALDI-TOF MS, and EDX, was instrumental for describing and characterising these two new *Syncephalastrum* species.

## Figures and Tables

**Figure 1 jof-10-00064-f001:**
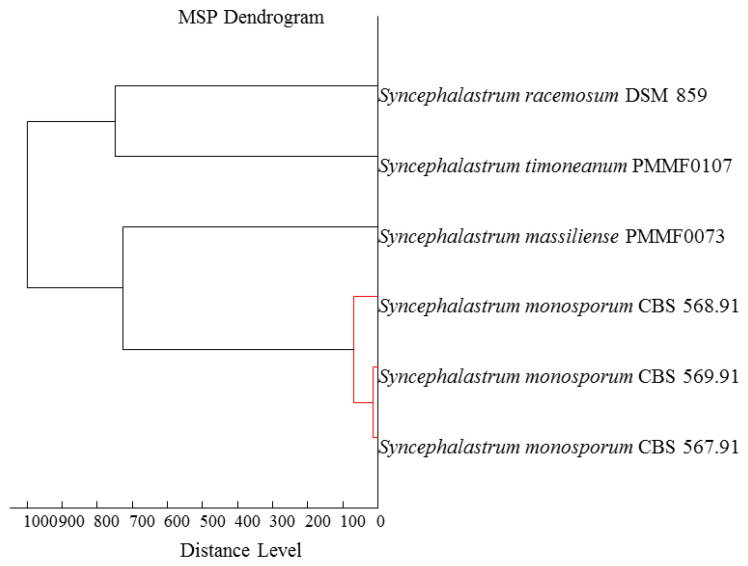
MALDI-TOF MS dendrogram based on the protein expression intensity profile for the four reference strains and two new species of *Syncephalastrum* generated using the MALDI-TOF Biotyper Compass Explorer software V.4.1 (Bruker Daltonics).

**Figure 2 jof-10-00064-f002:**
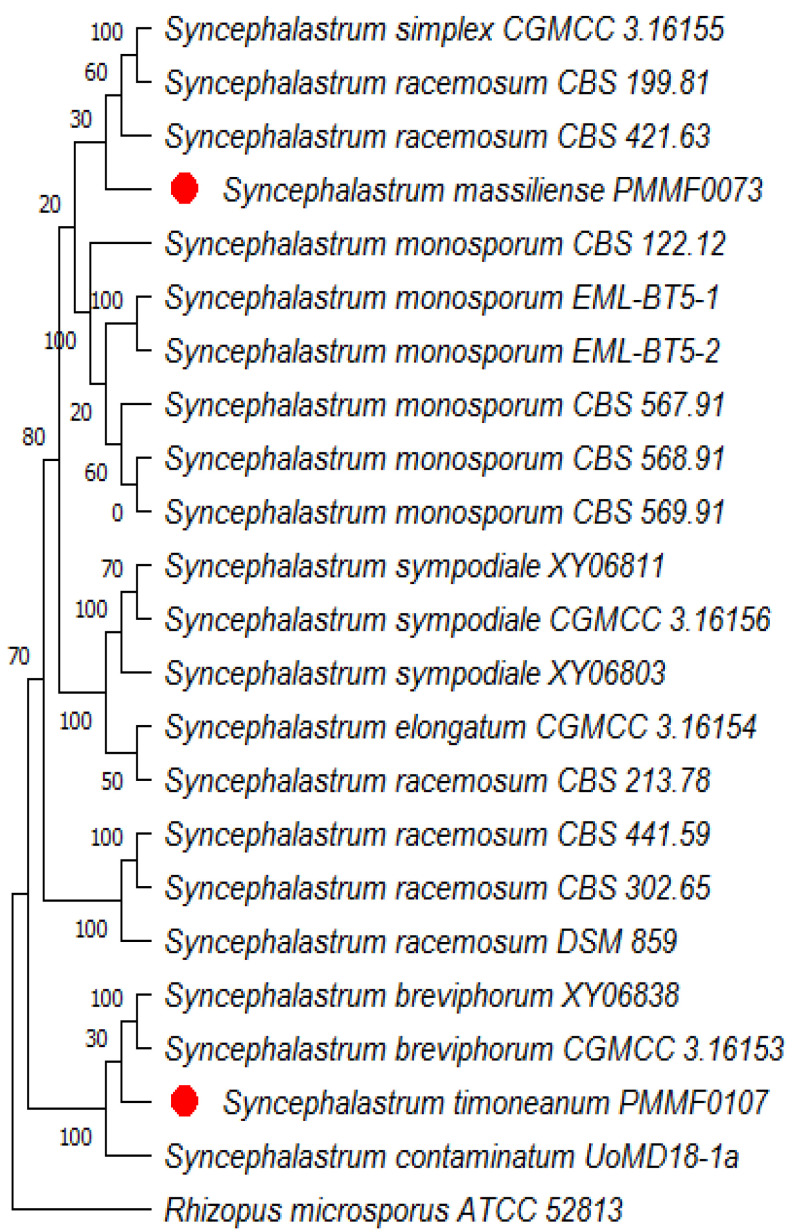
Maximum parsimony dendrogram based on the ITS sequences, including *Syncephalastrum massiliense* PMMF0073, *Syncephalastrum timoneanum* PMMF0107, and 20 other *Syncephalastrum* spp. reference strains. *Rhizopus microsporus* ATCC 52813 was used as the outgroup. The tree was constructed via the maximum parsimony method using MEGA 11 software. The bootstrap values were estimated at 1000 replications.

**Figure 3 jof-10-00064-f003:**
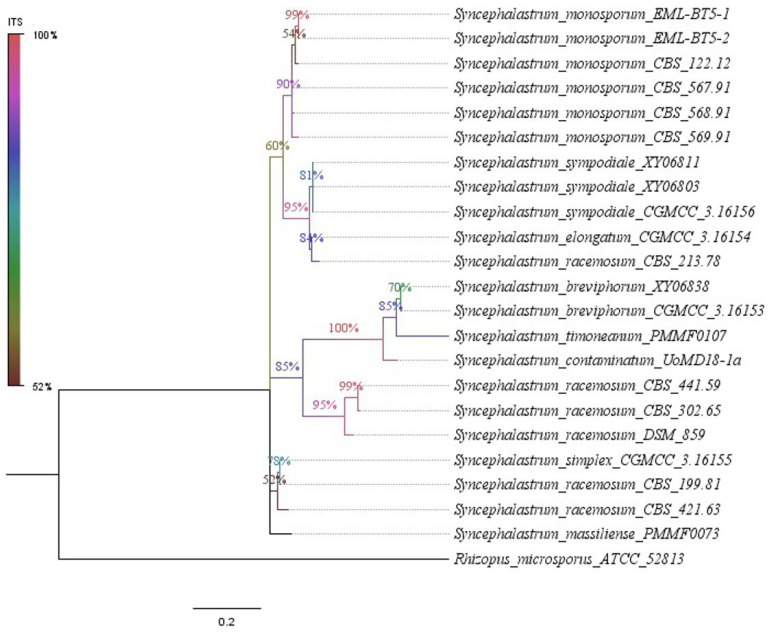
Bayesian phylogenetic tree based on the ITS sequences of *Syncephalastrum massiliense* PMMF0073, *S. timoneanum* PMMF0107, and 20 *Syncephalastrum* spp. reference strains. *Rhizopus microsporus* ATCC 52813 was used as the outgroup. The tree was constructed using the MrBayes software (3.2.7a) and Figtree (V.1.4.4).

**Figure 4 jof-10-00064-f004:**
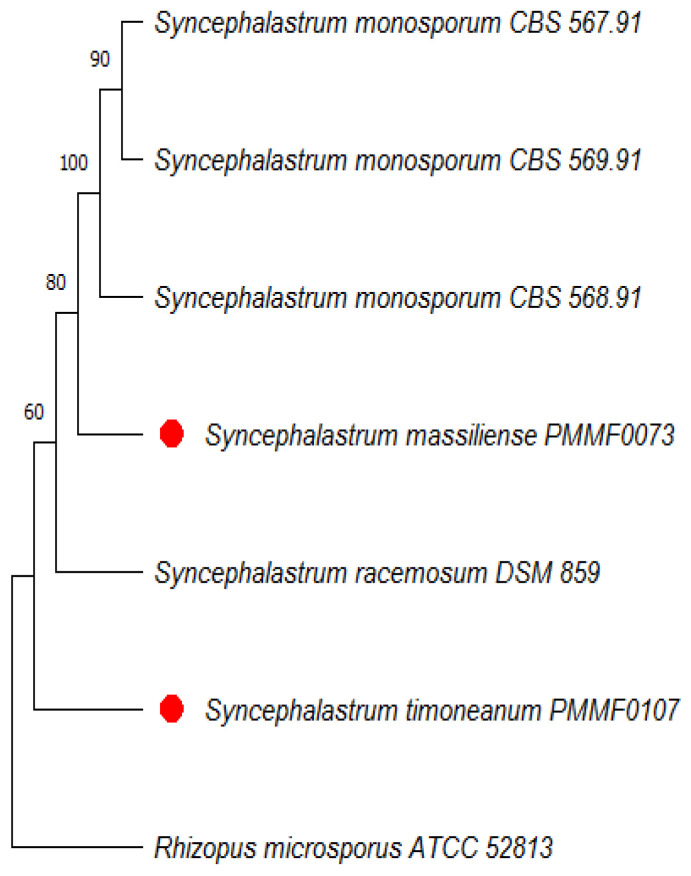
Maximum parsimony dendrogram based on the concatenated ITS, TUB2, TEF1, and D1/D2 sequences of *Syncephalastrum massiliense* PMMF0073, *S. timoneanum* PMMF0107, and four *Syncephalastrum* spp. type strains. *Rhizopus microsporus* ATCC 52813 was used as the outgroup. The tree was constructed via the maximum parsimony method using the MEGA 11 software. The bootstrap values were estimated at 1000 replications.

**Figure 5 jof-10-00064-f005:**
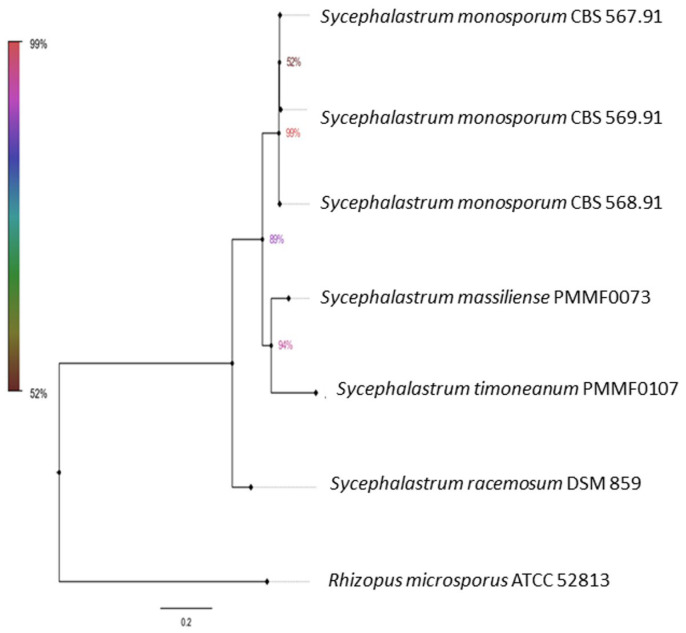
Bayesian phylogenetic tree based on the concatenated ITS, TUB2, TEF1, and D1/D2 sequences of *Syncephalastrum massiliense* PMMF0073, *S. timoneanum* PMMF0107, and four *Syncephalastrum* spp. type strains. *Rhizopus microsporus* ATCC 52813 was used as the outgroup. The tree was constructed using the MrBayes software (3.2.7a) and Figtree (V.1.4.4).

**Figure 6 jof-10-00064-f006:**
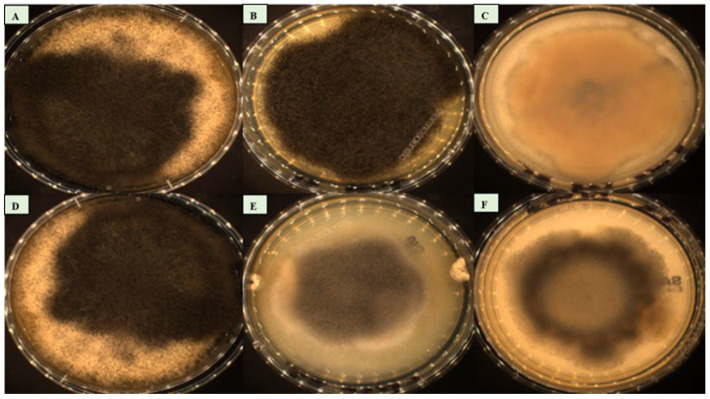
Culture growth on the SDA GC medium at 25 °C. (**A**) *Syncephalastrum massiliense* PMMF0073. (**B**) *Syncephalastrum timoneanum* PMMF0107. (**C**) *S. racemosum* DSM 859. (**D**) *S. monosporum* CBS 567.91. (**E**) *S. monosporum* CBS 568.91. (**F**) *S. monosporum* CBS 569.91.

**Figure 14 jof-10-00064-f014:**
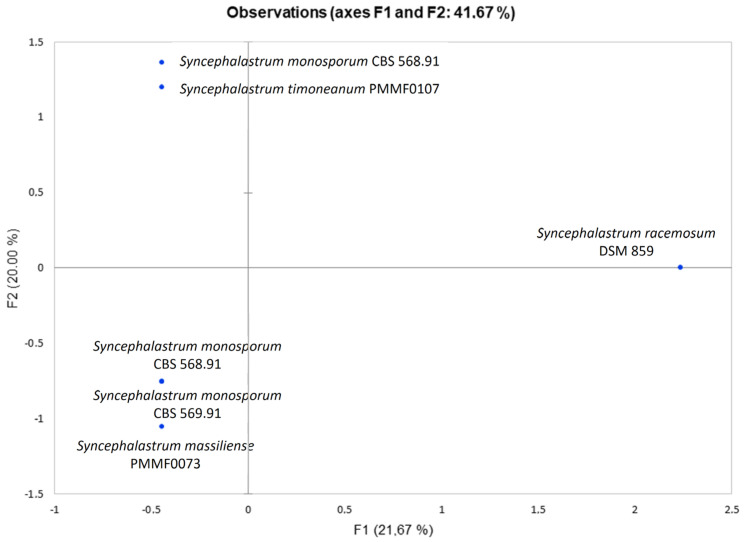
Principal component analysis of the EDX (energy-dispersive X-ray spectroscopy) chemical mapping profile for the four reference strains and the two novel *Syncephalastrum* species. In this analysis, computed using the XLSTAT software V.2022.4.1, the principal components F1 and F2 explained 41.7% of the EDX data variance.

**Figure 15 jof-10-00064-f015:**
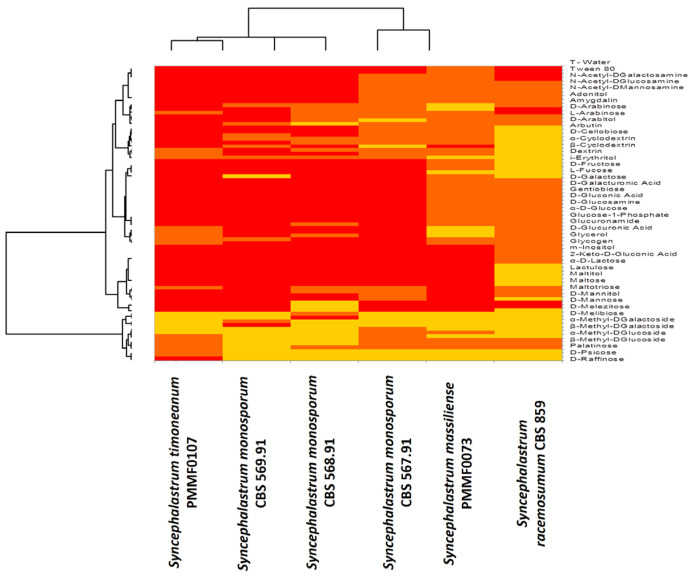
Heat map of the carbon sources assimilation assessed using the Biolog™ system for the four reference strains and two species of *Syncephalastrum*, computed using the XLSTAT software V.2022.4.1. Colour gradient interpretation: the most assimilated substrates are in red, and the least assimilated substrates are in yellow.

**Table 1 jof-10-00064-t001:** Panel of the primers used for amplifying the ITS, TUB2, TEF-1 alpha, and D1/D2 genetic regions.

Primers	Sequences	Targeted Regions	References
ITS1	TCCGTAGGTGAACCTGCGG	18S–5.8S	[27]
ITS2	GCTGCGTTCTTCATCGATGC	18S–5.8S	[27]
ITS3	GCATCGATGAAGAACGCAGC	5.8S–28S	[27]
ITS4	TCCTCCGCTTATTGATATGC	5.8S–28S	[27]
ITS1	TCCGTAGGTGAACCTGCGG	18S–5.8S, 5.8S–28S	[27]
ITS4	TCCTCCGCTTATTGATATGC	18S–5.8S, 5.8S–28S	[27]
Bt-2a	GGTAACCAAATCGGTGCTGCTTTC	TUB2	[28]
Bt-2b	ACCCTCAGTGTAGTGACCCTTGGC	TUB2	[28]
EF1-728F	CATCGAGAAGTTCGAGAAGG	TEF1	[29]
EF1-986R	TACTTGAAGGAACCCTTACC	TEF1	[29]
D1	AACTTAAGCATATCAATAAGCGGAGGA	28S	[30]
D2	GGT CCG TGT TTC AAG ACG G	28S	[30]

**Table 2 jof-10-00064-t002:** Fungal collection strain ID and the GenBank accession numbers of the nucleotide sequences used in the phylogenetic analysis.

Species	Strain ID	GenBank Accession Numbers			
		ITS	TUB2	D1/D2	TEF1
*Syncephalastrum massiliense*	PMMF0073	OL699905.1	ON149883	OM417069.1	OM362516.1
*Syncephalastrum timoneanum*	PMMF0107	OL699906.1	ON149884	OM417070.1	OM362517.1
*Syncephalastrum racemosum*	DSM 859	OL699907.1	ON149885	OM417071.1	OM362518.1
*Syncephalastrum monosporum*	CBS 567.91	OL699908.1	ON149886	OM417072.1	OM362519.1
*Syncephalastrum monosporum*	CBS 568.91	OL699909.1	ON149887	OM417073.1	OM362520.1
*Syncephalastrum monosporum*	CBS 569.91	OL699910.1	ON149888	OM417074.1	OM362521.1
*Syncephalastrum monosporum*	CBS 122.12	HM999977.1	NA	JN206575.1	NA
*Syncephalastrum racemosum*	CBS 441.59	HM999985.1	NA	MH869451.1	NA
*Syncephalastrum racemosum*	CBS 302.65	HM999984.1	NA	MH870214.1	NA
*Syncephalastrum racemosum*	CBS 213.78	HM999978.1	NA	MH872886.1	NA
*Syncephalastrum racemosum*	CBS 421.63	HM999973.1	NA	MH869932.1	NA
*Syncephalastrum racemosum*	CBS 199.81	HM999972.1	NA	HM849718.1	NA
*Syncephalastrum racemosum*	EML-BT5-1	KY047152.1	NA	KY047158.1	NA
*Syncephalastrum racemosum*	EML-BT5-2	KY047143.1	NA	KY047157.1	NA
*Syncephalastrum simplex*	CGMCC 3.16155	OL678220	NA	NA	NA
*Syncephalastrum sympodiale*	XY06811	OL678223	NA	NA	NA
*Syncephalastrum sympodiale*	XY06803	OL678222	NA	NA	NA
*Syncephalastrum sympodiale*	CGMCC 3.16156	OL678221	NA	NA	NA
*Syncephalastrum elongatum*	CGMCC 3.16154	OL678219	NA	NA	NA
*Syncephalastrum breviphorum*	XY06838	OL678218	NA	NA	NA
*Syncephalastrum breviphorum*	CGMCC 3.16153	OL678217	NA	NA	NA
*Syncephalastrum contaminatum*	UoMD18-1a	MK799842	NA	NA	NA
*Rhizopus microsporus*	ATCC 52813	KU358721	XM_023614	Km1037773	XM_0236122

NA, not available.

**Table 3 jof-10-00064-t003:** E-test minimal inhibitory concentrations (MICs) of ten antifungal drugs against six refrence *Syncephalastrum* strains, including the two novel species: *S. massiliense*, and *S. timoneanum*.

MIC (mg/L)	AMB	AND	CAS	MIC	5-FC	FL	ITC	POS	VOR	IS
*Syncephalastrum massiliense* PMMF0073	0.125	>32	>32	>32	>32	>256	4	>32	>32	>32
*Syncephalastrum timoneanum* PMMF0107	0.047	>32	>32	>32	>32	>256	1	6	>32	>32
*Syncephalastrum racemosum* DSM 859	0.25	>32	>32	>32	>32	>256	0.75	0.75	>32	>32
*S. monosporum* CBS 567.91	>32	>32	>32	>32	>32	>256	>32	>32	>32	>32
*S. monosporum* CBS 568.91	>32	>32	>32	>32	>32	>256	>32	>32	>32	>32
*S. monosporum* CBS 569.91	>32	>32	>32	>32	>32	>256	>32	>32	>32	>32

AMB, amphotericin B; AND, anidulafungin; CAS, caspofungin; MIC, micafungin; 5-FC, flucytosin; FL, fluconazole; ITC, itraconazole; POS, Posaconazole; VOR, voriconazole; and IS isavuconazole.

## Data Availability

The *Syncephalastrum massiliense* holotype is available in the IHU MI (No PMMF0073) and IHEM (No 28561) strain collections. The nucleotide sequences are available from GenBank (Accession Numbers: OL699905, ON149883, OM362516, and OM417069). The datasets analysed during the current study are available from the corresponding authors upon reasonable request. The *Syncephalastrum timoneanum* holotype is available in the IHU MI (No PMMF0107) and IHEM (No 28562) strain collections. The nucleotide sequences are available from GenBank (Accession Numbers: OL699906, ON149884, OM362517 and OM417070).

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
