# Peer review of "Syncephalastrum massiliense sp. nov. and Syncephalastrum timoneanum sp. nov. Isolated from Clinical Samples"

_jof, 2024, doi:10.3390/jof10010064_

Round 1
Reviewer 1 Report (Previous Reviewer 1)
Comments and Suggestions for Authors
Dear Authors,
Although I acknowledge the importance of uncovering undersampled fungal diversity, describing new species is governed by its own rules and requirements resulting from the nomenclature code. There are several species concepts as well as species delimitatin and identification methods. These are not synonimes. In general, it is rather bad practice to describe new species based on single strain! In such a case you are not able to measure intraspecific variability. My biggest reservations concern the presented phylogenetic trees, because S. racemosus is paraphyletic there. Therefore, your conclusions based on them are injustified.
Abstract:
I would like to have the crucial species specific differences already in the abstract. Not only statement "have a similar morphology to S. racemosum, but each display distinct phenotypic and genotypic features." Please, mention these specific features.
line 21: Did you analyse DNA sequence coding for RNA or RNA sequence? This is not clear from this sentence.
Introduction:
line 47: Better to use e.g. or such as instead of double bracket.
line 50: Does this sentence refer to the most common Synecephalastrum causing mucormycosis or it is general sententence. If it is the general one better to transfer it to the next paragraph. If it is about mucormycosis, please refrese it to be more specific.
line 54: Citation to previous paragraph.
line 61: without "spp"
lines 63-65: The presence of rhizoids is bad example as there are Mucor taxa forming rhizoids!
lines 104-105: varieties (var.) with a lowercase letters
Results:
Figure 2. is not supporting delimitation of Syncephalastrum massiliense and Syncephalastrum timoneanum as new species because S. racemosum is paraphyletic! Also the outgroup is incorrect as it is too distant!
Figure 3. Why only one strain per species is included in the tree. How the Authors want to asses intraspecific variation in this case?
Figure 4. S. racemosum is paraphyletic!
Figure 9. Intraspecific variability must be measured to plot differences between species.
Discussion:
I miss the critical discussion about species delimitation concepts in Mucorales or at least Syncephalastrum.
Several citations are still in wrong format.
Author Response
Although I acknowledge the importance of uncovering undersampled fungal diversity, describing new species is governed by its own rules and requirements resulting from the nomenclature code. There are several species concepts as well as species delimitatin and identification methods. These are not synonimes. In general, it is rather bad practice to describe new species based on single strain! In such a case you are not able to measure intraspecific variability. My biggest reservations concern the presented phylogenetic trees, because S. racemosus is paraphyletic there. Therefore, your conclusions based on them are injustified.
Answer: We agree that species delimitation and identification methods are distinct but they also depend on each other. We also agree that it is preferable to describe a new species on more than only one strain. But this attitude severely limits the description of new taxa and increases fungal species heterogeneity and intraspecific variability, notably in nucleotide databases, which in turns hampers the species delimitation. We acknowledge that the nucleotide sequences of the S. racemosum strains are heterogeneous. Noteworthy, some of them have been previously considered as separate species or subspecies. Yet I acknowledge that it remains unclear for me why the reviewer considers S. racemosum paraphyletic. Phylogeny and taxonomy are complex and have always be a matter of dispute among specialists. We believe in our conclusions mainly because of the low percentage of nucleotide sequence identity between each of the two new species and each of the type strains of the previously described species in the Syncephalastraceae.
Abstract:
I would like to have the crucial species specific differences already in the abstract. Not only statement "have a similar morphology to S. racemosum, but each display distinct phenotypic and genotypic features." Please, mention these specific features.
Answer: The main characteristic features have been added in the abstract.
line 21: Did you analyse DNA sequence coding for RNA or RNA sequence? This is not clear from this sentence.
Answer: We analyzed the DNA sequence coding for RNA. This has been clarified in the sentence.
Introduction:
line 47: Better to use e.g. or such as instead of double bracket.
Answer: corrected.
line 50: Does this sentence refer to the most common Synecephalastrum causing mucormycosis or it is general sententence. If it is the general one better to transfer it to the next paragraph. If it is about mucormycosis, please refrese it to be more specific.
Answer: The sentence was about mucormycosis. It has been clarified.
line 54: Citation to previous paragraph.
Answer: Sorry, it has been corrected this time.
line 61: without "spp"
Answer: corrected.
lines 63-65: The presence of rhizoids is bad example as there are Mucor taxa forming rhizoids!
Answer: We agree, it has been corrected this time.
lines 104-105: varieties (var.) with a lowercase letters
Answer: corrected.
Results:
Figure 2. is not supporting delimitation of Syncephalastrum massiliense and Syncephalastrum timoneanum as new species because S. racemosum is paraphyletic! Also the outgroup is incorrect as it is too distant!
Answer: The outgroup has been changed, as suggested.
In figure 2, S. timoneanum is not clustering within the S. racemosum clade, but with the S. monosporum clade. Even if S. racemosum is paraphyletic, S. massilisense and S. timoneanum show, for each of the four genes, percentages of identity much lesser than 99%, which is the usual threshold for pretending to a good identification at the species level.
The results of the BLAST with NCBI are presented below. For more reliability, we selected the three following parameters: Models (XM/XP), Uncultured/environmental sample sequences and Sequences from type material.
Figure 3. Why only one strain per species is included in the tree. How the Authors want to asses intraspecific variation in this case?
Answer: The tree (Figure 3) was constructed with the concatenated sequences of ITS, TUB2, TEF1 and D1/D2. We were not able to include more NCBI strains, since the sequences for the four loci were unavailable for each of them. We could only consider strains for which the sequences of the four loci were available.
Figure 4. S. racemosum is paraphyletic!
Answer: Sorry, but the paraphyletic position of S. racemosum remains unclear to us, based on our data. And if this should be, would it prevent us from describing new taxa?
Figure 9. Intraspecific variability must be measured to plot differences between species.
Answer: We included only the type strains of each species in thus analysis. This was because of the feasibility but also chiefly to avoid heterogeneity due to a potential misidentification/classification bias. As there is only one strain per species, the intraspecific variability cannot be assessed.
Discussion:
I miss the critical discussion about species delimitation concepts in Mucorales or at least Syncephalastrum.
Answer: Wed agree, this missing part have been added to the text.
Several citations are still in wrong format.
Answer: Sorry, we have corrected the citations format.
Reviewer 2 Report (Previous Reviewer 2)
Comments and Suggestions for Authors The manuscript is well written in edited format.Author Response
We thank the reviewer for his kind words of encouragement
This manuscript is a resubmission of an earlier submission. The following is a list of the peer review reports and author responses from that submission.
Round 1
Reviewer 1 Report
Comments and Suggestions for Authors
Presented manuscript aims to describe two new species of Syncephalastrum. Basal fungal lineages are generally understudied, so each study describing new taxa is important. However, the presented manuscript needs significant improvements before publishing. The main problem is that the phylogenetic trees described in the paper are not included in the manuscript. Without them, it is hard to understand the presentation of results. For example, it is not clear to me if this species description is based on only one strain or there were some more also included. There are also several misunderstandings and misinterpretations in the introduction. I put some specific comments in the manuscript. I really appreciate the polyphasic approach adopted by Authors but it would be also good to include some comments on intra and interspecific variation within the Syncephalastrum. Finally, I think that the paper can be reconsidered after major revisions.

Reviewer 2 Report
Comments and Suggestions for Authors
Authors try to describe Syncephalastrum massiliense sp. nov. and Syncephalastrum timoneanum sp. nov., isolated from clinical samples.
1. line 10 “Mucormycosis is known to be a rare opportunistic infection caused by Syncephalastrum species”. Mucormycosis refers to infection caused by diverse fungal organisms in the order Mucorales, including those in the genera Rhizopus, Rhizomucor, and Mucor. not only Syncephalastrum species. Please change the sentence.
2. How do you confirm that these infections caused by those species? How many times did you take samples from the patient and were the same answers repeated in different samplings? How did you differentiate between colonization and disease?
3. References should be numeric.
4. Line 35-36 it is better to write “with low pathogenicity and are rarely known to cause human diseases”
5. All geniuses of fungi should be written in italic format.
6. Why didn't you use micro dilution method for anti-fungal susceptibility testing?
7. In discussion section you should also discuss about anti-fungal susceptibility, and overall you should extend discussion section.
Reviewer 3 Report
Comments and Suggestions for Authors
Dear Authors,
In this form, the manuscript is not acceptable. Please check the current taxonomy of Mucoromycota, please check the reference list. The strain names must be in italic. Figures 1, 2, 3, and 4 are missing from the manuscript. The resolution of fig 5 is low.